# Policies to Create Healthier Food Environments in Canada: Experts’ Evaluation and Prioritized Actions Using the Healthy Food Environment Policy Index (Food-EPI)

**DOI:** 10.3390/ijerph16224473

**Published:** 2019-11-14

**Authors:** Lana Vanderlee, Sahar Goorang, Kimiya Karbasy, Stefanie Vandevijvere, Mary R L’Abbé

**Affiliations:** 1Department of Nutritional Sciences, University of Toronto, Toronto, ON M5S 1A8, Canada; sahar.goorang@utoronto.ca (S.G.); kimiya.karbasy@utoronto.ca (K.K.); mary.labbe@utoronto.ca (M.R.L.); 2School of Public Health and Health Systems, University of Waterloo, Waterloo, ON N2L 3G1, Canada; 3School of Population Health, The University of Auckland, Auckland 1142, New Zealand; Stefanie.Vandevijvere@sciensano.be; 4Department of Epidemiology and Public Health, Sciensano (Scientific Institute of Public Health), 1050 Brussels, Belgium

**Keywords:** food environment, food policy, health policy, nutrition

## Abstract

Food environment policies play a critical role in shaping food choices, diets, and health outcomes. This study endeavored to characterize and evaluate the current food environment policies in Canada using the Healthy Food Environment Policy Index (Food-EPI) to compare policies in place or under development in Canada as of 1 January 2017 to the most promising practices internationally. Evidence of policy implementation from the federal, provincial, and territorial governments was collated and verified by government stakeholders for 47 good practice indicators across 13 policy and infrastructure support domains. Canadian policies were rated by 71 experts from across Canada, and an aggregate score of national and subnational policies was created. Potential policy actions were identified and prioritized. Canadian governments scored ‘high’ compared to best practices for 3 indicators, ‘moderate’ for 14 indicators, ‘low’ for 25 indicators, and ‘very little or none’ for 4 indicators. Six policy and eight infrastructure support actions were prioritized as the most important and achievable. The Food-EPI identified some progress and considerable gaps in policy implementation in Canada, and highlights a particular need for greater attention to prioritized policies that can help to shift to a health-promoting food environment.

## 1. Introduction

It is well established that food choices are heavily influenced by the food environment within which they are made [1]. The food environment, broadly defined as the physical, economic, political, and sociocultural factors that influence dietary choices, is closely related to diet [2]. The types and quality of foods in the food supply, the cost, and the marketing of foods all play critical roles in shaping food choices, with an ultimate influence on overall health [3]. Current food environments are largely not conducive to supporting healthier eating behaviors [1,4]: Nutrition-related risk factors are now the greatest contributor to disability-adjusted life years and mortality worldwide [5].

Canada is no exception to this global trend. Overall diet quality in Canada is poor, with few Canadians meeting recommendations for fruit and vegetable consumption and sodium intake [6,7,8]. In Canada, diet-related conditions, including obesity and non-communicable diseases (NCDs) have considerable economic and societal impacts, estimated at approximately $13 billion CAD per year [9]. Current Canadian food environments are similar to global food environments in that they do not support the consumption of healthy, sustainable diets [1,4,10]. For example, as of 2017, only 14% of Canadian foods met Health Canada’s targets for sodium reduction, with nearly half showing no progress in meeting established voluntary sodium targets between 2012 and 2016 [11]. Like most countries, the marketing of unhealthy foods in Canada is common, particularly targeted to children. An online survey of primary and secondary schools in three Canadian provinces found various types of marketing to children in 84% of schools [12], and marketing to children via digital media and television is pervasive [13,14]. And while the majority of Canadians report noticing and using nutrition information on food labels [15,16], consumers’ ability to interpret the information provided to them is limited [17,18,19,20], and there is currently no standardized interpretive information on the front of food packages in Canada.

Governments play an important role in shaping the food environment, as policies have the potential to shift the environment towards one that promotes healthy diets. Globally, many middle and high-income countries are assuming leadership in implementing food environment policies, regulations, and programs at the national, state, and local level [21]. In Canada, the announcement of a Healthy Eating Strategy [22] and A Food Policy for Canada [23] (a national food strategy) in 2016 brought food environment policy to the forefront of policy discussions. The interest and attention on food environments in Canada provided a unique opportunity to evaluate the food policy environment. The objective of the current study was to comprehensively evaluate policies and actions that federal, provincial, and territorial governments in Canada are taking to create healthy food environments, and to prioritize areas for action to address identified current policy gaps, using the Healthy Food Environment Policy Index (Food-EPI).

## 2. Materials and Methods

This study employed the Healthy Food Environment Policy Index (Food-EPI) tool and process developed by the International Network for Food and Obesity Research, Monitoring and Action Support (INFORMAS) to assess Canadian food environment policies compared to best practice [3]. All participants gave their informed consent for inclusion before they participated in the study. The study was conducted in accordance with the Declaration of Helsinki, and the protocol was approved by the University of Toronto Human Research Ethics board (Protocol ID 33249).

The Food-EPI tool and process are described in detail in previous publications [24,25]. Briefly, the Food-EPI assesses government actions within two primary components: (1) policy and (2) infrastructure support. Within these components, 7 policy ‘domains’ (food composition, food labelling, food promotion, food prices, food provision, food retail, and food trade and investment) and 6 infrastructure support ‘domains’ (leadership, governance, monitoring and intelligence, funding and resources, platforms for interaction, and health-in-all-policies) are assessed, using a set of 47 ‘good practice indicators’ for specific policy areas relevant to each domain. For each indicator, a set of international benchmark examples have been compiled to illustrate promising practices whereby governments (at any level) have taken action or implemented policy. The scope of Food-EPI is focused on obesity and diet-related NCDs, and at the present time, does not include other policy areas relevant to nutrition such as genetically modified organisms, food safety, food production, food security, undernutrition, micronutrient deficiencies, and breastfeeding or infant formula, and environmental/climate change policies. The full set of indicators of the Food-EPI can be found in Appendix A, as well as the government levels (federal and/or provincial/territorial) that have at least some regulatory responsibility for each indicator in Canada.

### 2.1. Adaptation of the Food-EPI to the Canadian Context

Given the nature of regulatory jurisdiction in Canada, food environment policies and related infrastructure support were evaluated for federal as well as provincial/territorial governments. Out of the 47 good practice indicators under evaluation, 7 were identified as outside of the regulatory authority of provincial and territorial governments in Canada (composition targets for processed foods, nutrition information on food labels, health claim regulations, front of package food labelling, incorporation of risk impact assessments in trade and investment agreements, measures to manage and protect regulatory capacity, and population-level food-based dietary guidelines) and were only rated at the federal level. The indicator for a systems-based approach with local organizations was erroneously excluded from provincial/territorial ratings as the result of a computer programming error, and was only analyzed at the federal level. One indicator was identified as not relevant at the federal level (support for implementation of nutrition policies in school settings) and was only rated at the provincial/territorial level.

### 2.2. Policy Search, Evidence Document Development, and Policy Verification

Evidence on implementation of food environment policies and actions in Canada was collated up to 1 January 2017. Data were compiled through comprehensive online searches of government websites and databases, policy documents, and non-governmental and advocate websites. The evidence documents were then shared with government stakeholders in relevant ministries to verify that the information collated was correct and comprehensive, to the extent possible, including Health Canada (Ministry of Health), the Public Health Agency of Canada, Department of Finance, Finances Canada, Global Affairs Canada, and others. The final evidence documents for each of the provinces and territories and the federal government are available at: http://labbelab.utoronto.ca/food-epi-canada-2017/.

### 2.3. Expert Panel

A panel was convened from the various areas of food and nutrition research, practice, and advocacy in Canada. A total of 111 experts were invited, 75 from academia and 36 from non-governmental or other civil society organizations. The group included academic experts, health and nutrition-related non-governmental organizations, and individuals from other types of organizations with relevant expertise and with no conflict of interest in participating. Potential participants were identified through the authors’ extensive networks, and snowball sampling was conducted among government stakeholders to identify other potential experts to invite to participate in the process. Individuals with government or industry affiliations were purposefully excluded from the sample to avoid conflict of interest in rating policy actions; however, key government stakeholders were invited to be observers and attended all of the rating workshops.

### 2.4. Rating Process

A two-step process was used to rate federal and provincial/territorial policies in Canada in May and June of 2017. First, each expert was randomized to rate 1 of 12 provinces or territories (note that the territory of Nunavut was excluded from the present evaluation, after careful consideration of Nunavut’s unique food environment [26,27]) using an online process. For federal (national) policies, ratings were conducted in three in-person workshops in Toronto (*n* = 2) and Vancouver (*n* = 1).

Experts were asked to review the relevant evidence documents for the provincial/territorial jurisdiction to which they were randomized prior to undertaking the online process and for the federal jurisdiction prior to the workshop. The process for online and in-person workshop ratings was similar. For each good practice indicator, the international benchmark examples were presented to identify what types of policies and regulations had been implemented in other jurisdictions. Next, a brief paragraph of context for the Canadian policy environment was provided, and the current evidence of implementation by the federal/provincial/territorial government for the indicator was described in detail. An example of the data that were provided in the online survey is depicted in Appendix A.

Participants were asked to rate Canadian federal/provincial/territorial policies in comparison to the international benchmarks that were presented on a 5-point scale (0–20%, 21–40%, 41–60%, 61–80%, 81–100% implementation compared to best practice). Participants were instructed to rate the level of implementation in Canada compared to the international benchmarks, and not against theoretical or ideal standards. There were instances where the international benchmarks did not cover all aspects of the good practice indicator; however, the process purposefully compared Canadian policies to real-world examples of policy action, despite some examples being limited in scope compared to the policy described in the theoretical good practice statement and some examples of implemented policies being limited in their strength or comprehensiveness.

Participants were asked to consider the quality of the policies compared to international best practice and the extent of implementation, bearing in mind the various levels of the policy cycle (agenda setting and initiation, policy development, implementation, and evaluation), and ratings were to be assigned accordingly. For example, policies in the policy development phase would be rated lower than policies in the implementation or evaluation stages. Implementation was considered to include public intentions and plans of the government, as well as government funding for implementation of actions taken by NGOs and working or advisory groups.

### 2.5. Identification of Proposed Actions

After the ratings workshops were conducted, based on implementation gaps identified, a long list of federal policy and infrastructure support actions (*n* = 60) was compiled using the feedback from all three workshops to identify concrete actions that could be recommended to the government to improve the Canadian food environment. Unique provincial and territorial actions were created based on relevant ratings and federal policy discussions. The long list of actions was cross-referenced with proposed policy actions and policy positions from prominent non-governmental organizations and groups that were participating in the Food-EPI Canada process, including Alberta Policy Coalition for Chronic Disease Prevention [28], the Canadian Hypertension Advisory Committee [29], Dietitians of Canada [30], the Stop Marketing to Kids Coalition [31], Heart and Stroke Foundation [32], and the Canadian Medical Association [33] to examine alignment with current policy recommendations from other groups.

The long list of all proposed actions was circulated to all workshop participants, and participants were given an opportunity to provide feedback with regards to wording, content, and scope of the proposed actions, as well as identify any policy actions that may have been missed in the summary. This feedback was incorporated, to the extent possible, in the development of the final proposed policy and infrastructure support actions, while considering the sentiments of the larger group and discussion dialogues.

### 2.6. Prioritization Ranking of Proposed Actions

In the final activity, participants were asked to rate the policy and infrastructure support actions according to two elements: ‘Importance’ and ‘Achievability’. Importance included criteria such as the need, impact, effect on equity, and other positive and negative effects of the policy/action. Achievability included criteria such as the feasibility, acceptability, affordability, and efficiency of the policy/action.

Microsoft Excel worksheets were compiled with all of the proposed policy and infrastructure support actions for the federal and provincial/territorial proposed actions. For the federal policies, experts were given an overall allotment of 150 points for ‘Importance’ and 150 points for ‘Achievability’ to be allocated to the 30 proposed policy actions (all scores were initially set to 5 points each). Policy actions that were considered more important or achievable scored more points, while lower priority actions scored fewer. Similarly, participants were asked to rate the importance and achievability of the 30 proposed infrastructure support actions, with 150 points for importance and 150 points for achievability. The same exercise was completed for each province and territory for which the expert had been randomized. The total number of points for each province or territory varied by the number of proposed actions. Participants were also given an opportunity to weight the ‘Importance’ and ‘Achievability’ criteria, to indicate whether the expert placed greater value on the importance or achievability of a policy. Weighting was initially set at 50%–50% for ‘Importance’ and ‘Achievability’.

### 2.7. Evaluation of the Food-EPI Canada Process

A process evaluation was conducted to examine expert evaluations of participation in the Food-EPI process. Participants were asked to rate the comprehensiveness of the Food-EPI tool and the comprehensiveness and completeness of the Federal Evidence document (1 = not sufficiently comprehensive, 5 = too comprehensive) and how easy it was to conduct the ratings in the workshop and the online ratings (1 = easy, 5 = difficult). Then, participants were asked their level of agreement with the following statements: “My knowledge of food environments and related food and nutrition policy increased as a result of the project”, “I have increased my knowledge of current best practice/what other governments are doing internationally”, “I made new professional connections or strengthened existing relationships as a result of the project”, “The project is likely to contribute to beneficial policy change” and “It is important to repeat the study in order to monitor government progress.” Lastly, participants were asked whether or not they would be willing to participate in the assessment process again if the Food-EPI were to be repeated.

## 3. Analysis

Inter-rater reliability was assessed using AgreeStat, estimated as the percentage of agreement between experts using quadratic weights (Agreestat 2013.1, Advanced Analytics, Gaithersburg, MD, USA). For the estimation of variance, the sample of subjects (i.e., Food-EPI indicators included) was set at 100%, and the sample of raters was set at 64% (response rate for ratings workshops).

Any policy indicators for which greater than 20% of participants selected ‘cannot rate’ were excluded from analysis, as it was assumed that either the policy evidence document or the international benchmark were not well understood by participants. This excluded one indicator (sufficient population nutrition budget) from the ratings, as 51% of participants selected ‘cannot rate’ due to the lack of public information on funding that is specific to diet-related NCDs, obesity or population nutrition, and a poorly defined international benchmark for this indicator.

Mean ratings were assessed for the federal and provincial/territorial indicators. To assess overall food environment policy implementation incorporating both federal and provincial/territorial governments, a composite score was derived for each indicator that fell within both federal and provincial/territorial jurisdiction (*n* = 39). The composite score weighted the federal government and each of the provincial and territorial governments equally (e.g., the average rating of the federal and all provincial and territorial governments). Scores were grouped into four levels of implementation: very little or no implementation (0–25%), low implementation (26–50%), moderate implementation (51–75%,) and high implementation (>75%).

For the federal prioritization exercise, sensitivity analysis was conducted to examine whether report outcomes varied by weighted or unweighted data, and to examine any changes to overall trends when outlier data were removed. Outliers, or disproportionately high scores from experts that therefore influenced the overall process outcome, were defined as the assignment of more than 20 points to any one policy. Two participants had scores that met this criteria and sensitivity analysis suggested that exclusion resulted in minor changes to the overall trends, and it was decided that these responses would be excluded to decrease the overall influence of any one expert on the overall process outcome. There were no noticeable differences when weighting was applied to the prioritization exercise, and thus the weighted priorities were used for analysis.

## 4. Results

A total of 71 experts participated in the ratings, 22 provided feedback on the proposed actions that were considered, including on the province or territory that they rated, any other provinces or territories for which they had policy familiarity and the proposed federal actions, and 74 participated in the prioritization exercise, for an overall response rate of 67%. The number of experts that rated each of the provinces and territories ranged from 5 to 7 experts, and a minimum of 3 experts conducted the prioritization activity for each province and territory. A total of 7 government observers attended the different federal workshops to listen to the expert panel discussion and provide clarity on current policy status, as required. Overall, 62% of the experts were from academia, 32% were from non-governmental organizations, and 6% were from other civil society organizations. There was geographic representation from all areas of Canada: 24% (*n* = 17) were from western provinces, 51% (*n* = 36) from Ontario, 13% (*n* = 9) from Quebec, 11% (*n* = 8) from the Atlantic provinces, and 1% (*n* = 1) from the territories. Experts came from a variety fields of research and practice, including dietetics, nutrition, public health, health policy, health economics, law, agriculture, sustainability, food security, and epidemiology, among others.

Gwet’s AC^2^ inter-rater reliability coefficient for the federal ratings was 0.63 (95%CI 0.61–0.66), which is considered relatively good agreement. Simple percent agreement was 89.6% (95% CI 88.9–90.3%) between raters across all indicators. The Gwet’s AC^2^ inter-rater reliability coefficient for provincial and territorial ratings ranged from 0.33 to 0.92, with average inter-rater reliability of 0.64, and 9 of 12 jurisdictions had a coefficient greater than 0.5. In some instances, provinces and territories with similar or identical policies were rated differently, as a result of varied expert interpretation and a small number of experts rating within each province and territory. To address this, manual adjustments were made to two indicators (Minimize taxes on healthy foods (PRICE1) and Public Access to Information (GOVERNANCE4) to ensure that provinces or territories that were clearly meeting the international benchmark were rated as such.

Figure 1 represents the composite scores of federal and provincial/territorial policies. Overall, implementation by Canadian governments scored as ‘high’ compared to best practice for 3 indicators (providing nutrition information on food labels, minimizing taxes on healthy foods, and public access to information), ‘moderate’ for 14 indicators, ‘low’ for 25 indicators, and ‘very little or none’ for 4 indicators (increase taxes on unhealthy foods, planning policies for healthy food outlets, food availability and promotion in restaurants, and a systems-based approach with local organizations). The highest rated Food-EPI policy domain was ‘Provision’ and the highest Food-EPI infrastructure support domain was ‘Governance’. The lowest rated Food-EPI policy domain was Retail, and the lowest rated infrastructure support domain was Health-In-All-Policies.

See Appendix A for additional detail of federal and provincial/territorial government scores independently. When federal government policies were considered independently of provincial and territorial actions, implementation for 7 of 45 indicators scored as ‘high’ compared to best practice, and 9 scored as ‘very little or none’. All of the provinces and territories were scored ‘high’ for minimizing taxes on healthy foods and public access to information. Very few provinces had policies related to restricting promotion of unhealthy food to children across any channels (with the exception of Quebec, who was an international benchmark), menu labelling, and all policies within the ‘Retail’ domain.

The full list of prioritized policy action recommendations for the federal government can be found in Table 1 and prioritized infrastructure support action recommendations for federal government can be found in Table 2. Graphs depicting the assigned scores for ‘Achievability’ and ‘Importance’ of federal government scores can be found in Figure 2 and Figure 3, as well as the Food-EPI indicators that each policy is related to. Policies that were listed in the upper quadrant (*n* = 6 for policy actions and *n* = 8 for infrastructure support actions) were considered highest priority.

Prioritized policy and infrastructure support recommendations for each province and territory are listed in detail in summary reports for each province that are publicly available [34]. Overall, the policy actions that were most consistent ranked as higher priority across the 12 provinces and territories are shown in Table 3.

### Process Evaluation Results

Overall, participants rated the comprehensiveness of the Food-EPI tool ‘about right’ (mean 3.2 out of 5 on the Likert scale), and similarly rated the comprehensiveness of the federal evidence document (3.3 out of 5). Participants found the ratings somewhat challenging, with online ratings being somewhat more challenging (mean = 3.1 for workshop ratings and 3.5 for online ratings). Overall, 84% of participants agreed or strongly agreed that they had increased their knowledge of food environments and related food and nutrition policy, 86% had increased their knowledge of current best practice and international government actions, and 73% had made new connections in the area of food environments. Overall, 77% agreed or strongly agreed that the project was likely to have a policy impact, and 91% agreed that it was important to repeat the same study over time to assess progress.

## 5. Discussion

Overall, the results suggest that there are few policy and infrastructure support areas where Canadian governments are reaching international benchmarks for promising practices in food environment policy. The level of implementation varied between domains, and also between national and subnational governmental bodies. Notably, there was one Canadian policy that was an international benchmark, the restrictions on marketing to children in Quebec. Infrastructure support indicators had higher ratings of implementation than policy indicators, particularly for the federal government compared to provincial and territorial governments. The prioritization exercise rated policy and infrastructure supports separately, and did not compare the importance and achievability between these two components; however, the results from the rating exercise suggest that there may be larger gaps in policy implementation in the Canadian setting compared to implementation of infrastructure supports, where ratings were overall higher.

The large geographic area and multiple levels of government (i.e., federal and provincial/territorial) that bear regulatory jurisdiction over food environment policies in Canada represented a unique challenge in assessing the overall Canadian food policy environment. The Food-EPI process identified challenges with identifying ‘who’ is responsible for taking action to address some of these areas where significant gaps remain in many or most jurisdictions. The current patchwork of policies across provinces and territories that are implemented in some policy domains represent inherent inequities in exposure to less healthy food environments between Canadians living in different jurisdictions, which are likely contributing to disparities in diet quality and rates of obesity and NCDs between provinces and regions [35]. Given the unique jurisdictional capacity in Canada, many of the proposed policy actions would need to be implemented by multiple sub-national governments in order to create an equitable environment for all Canadians.

### 5.1. Strengths and Limitations

Strengths of the Food-EPI Canada process include the use of internationally-developed methods created by leading experts in food environment policy that have been implemented in 23 countries to date [2]. The tool is adaptable, and was contextualized to the Canadian policy landscape to ensure that the evaluation was internally valid. Engagement with a large expert panel with a broad range of expertise in various areas of food policy from across the country and a variety of types of organizations also strengthens the conclusions from the panel. This is also an implicit limitation, as experts are likely to have policy expertise in one or two policy domains and may have introduced some level of individual bias in each individual prioritization exercise; however, it is likely that using group scores helped to minimize this individual influence. In addition, the Food-EPI process requires experts to apply knowledge and experience in food environment policy to conduct the ratings, which can lead to varied interpretation of both international benchmarks and policy implementation. The Food-EPI process compared Canadian policies to actual policy benchmarks that have been implemented globally, thus framing the policy recommendations as achievable for governments. The use of such benchmarks was identified as a challenge for experts, as the ‘best practice’ policies that have been implemented globally may not (yet) achieve a ‘gold standard’ for what might be an ideal policy action, and in some instances, international examples were extremely limited. Additionally, given that some of the policies identified as promising or identified as a ‘benchmark’ in this study have not been evaluated to identify their real-world impact, their use as benchmarks may not be ideal. However, the policies nominated as benchmarks were assessed by the INFORMAS team as being the most likely to have an impact on food environments according to their strength and comprehensiveness. This was a transparent process that engaged government throughout, but did not permit self-rating by government stakeholders. Other countries, such as Mexico and Thailand, have included governments in the Food-EPI rating process as a means to increase uptake of the final results, likely resulting in both benefits and challenges to the process [36,37]. While the government was extensively involved throughout the research process in Food-EPI, including in-person meetings with senior officials and involvement as observers, future consideration of government involvement is necessary for subsequent evaluations using the Food-EPI in Canada to ensure uptake of the results and to support policy development and implementation. Lastly, the scope of the Food-EPI process and tool at present is limited to nutrition issues related to obesity and non-communicable diseases, and does not assess other policy elements that are of interest to policymakers, such as environmental and climate change-related issues. Future work may consider including more broad aspects of the food system to identify synergistic policy opportunities.

### 5.2. Policy Implications

The Food-EPI Canada process supports a call to action for increased and sustained actions to improve food environments in Canada. The results were endorsed by several of the major NGOs involved in the process, including the Canadian Cancer Society, Canadian Nutrition Society, Canadian Obesity Network (now Obesity Canada), Childhood Obesity Foundation, Diabetes Canada, Dietitians of Canada, and several other provincial and territorial organizations. The initial reports [38] published were acknowledged by major government decision makers and influencers, including Canada’s Chief Public Health Office, the President of the Public Health Agency of Canada, the Assistant Deputy Minister of Health, and several provincial and territorial ministers of health, with in-person meetings held with senior officials in each of those agencies.

Some of the policy and infrastructure support actions that were recommended in this process are captured in the Healthy Eating Strategy and A Food Policy for Canada, including restrictions on marketing to children [39], and revisions to Canada’s food-based dietary guidelines [40], among others. Yet many of the recommendations suggest strengthening or extending the Healthy Eating Strategy to ensure that the policy strategy continues beyond election cycles and that it incorporates some of the vanguard policies that are gaining political and academic support worldwide. The policies that were prioritized were identified as most likely to fill the greatest gaps in the current environment and have the greatest impact, while still having high acceptability and feasibility. It is likely that the importance and achievability of some of these policies will change over time, as other policies are introduced, sociocultural norms and behaviors change, and political priorities shift.

Research comparing these results to other countries who have implemented the Food-EPI process using an aggregated, weighted score for policy and infrastructure support actions suggest that Canada performs moderately well, although more policy action is required to achieve the amount of progress demonstrated by some other governments globally [41]. Notably, no countries received a ‘high’ Food-EPI score, indicating that globally, few countries are implementing comprehensive policies and infrastructure supports the enable healthier food environments and support healthier food choices [41] Several of the prioritized policy and infrastructure support actions highlighted in Food-EPI Canada have also been highlighted by other countries as being the most important and achievable actions that governments can consider, in particular the recommendations to restricting advertising of unhealthy foods to children and implement a sugary drink tax. More than 40 jurisdictions in over 20 countries have implemented sugary drink taxes, and at least 8 countries have mandatory restrictions on the advertising of unhealthy foods to children via broadcast or non-broadcast media [21]. This demonstrates the policy momentum that is gaining in some of these policy areas that were once considered radical [42,43]. Reassessment of Canadian food environment policies after several years will demonstrate whether or not the federal government has successfully implemented these strategies, and whether or not this has spurred subsequent action among provincial and territorial governments.

Bringing together experts to systematically assess Canadian food environment policy has the potential to identify areas with broad expert recognition as important contributors to shaping a food environment that is health promoting, and also serves as an opportunity to build capacity among experts working in this field. The current study identified key policy areas and infrastructure support areas that experts project will contribute to healthier food environments at the federal, provincial, and territorial level, some of which have been proposed in Canada and some which have yet to be considered by governments. The results will help to shape priority areas for action by current and future governments in years to come. This study contributes to the efforts of INFORMAS to broadly characterize the global food environment.

## 6. Conclusions

The Food-EPI Canada process supports a call to action for increased and sustained actions to improve food environments in Canada. The current study identified key policy areas (including restrictions on marketing to children, targets for nutrients of concern, and an excise tax on sugary drinks) and infrastructure support areas (including intake targets and monitoring for nutrients of concern and vegetables and fruit, revisions to Canada’s Food Guide, and monitoring of the Healthy Eating Strategy), that experts project will contribute to healthier food environments at the federal, provincial, and territorial level, some of which have been proposed in Canada and some which have yet to be considered by governments. The results will help to shape priority areas for action by current and future governments in years to come. This study contributes to the efforts of INFORMAS to broadly characterize the global food environment.

## Figures and Tables

**Figure 1 ijerph-16-04473-f001:**
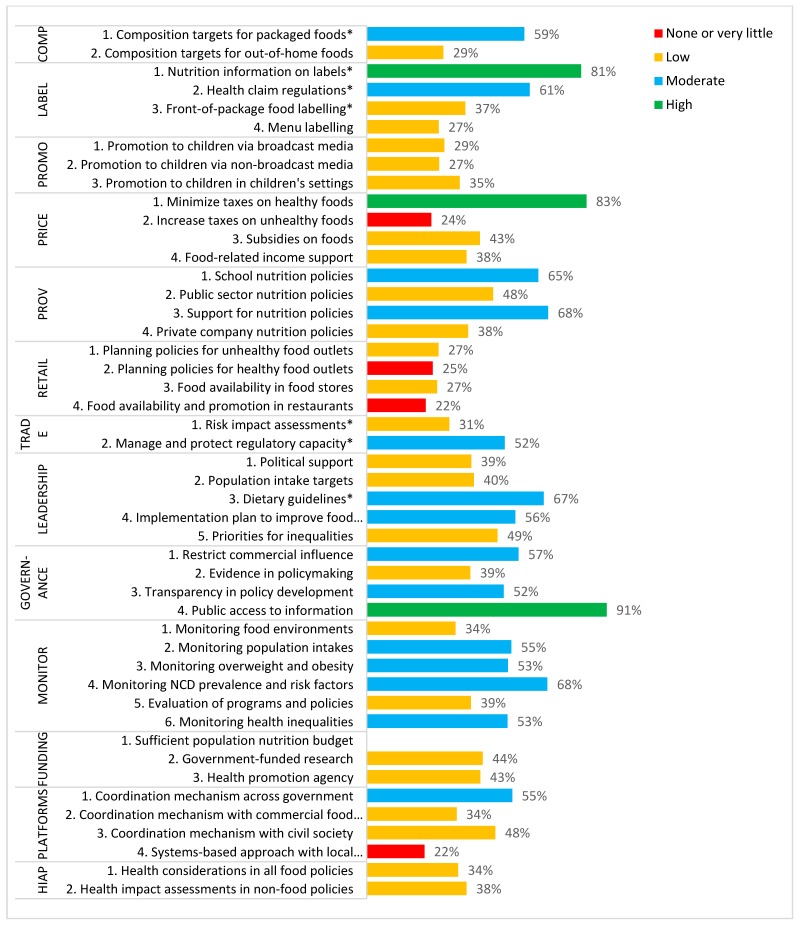
Composite score of federal and provincial/territorial ratings of the level of implementation compared to best practice for 46 Food Environment Policy Index (Food-EPI) indicators within 7 policy and 6 infrastructure support domains. * indicates indicators that were rated based on federal policies only.

**Figure 2 ijerph-16-04473-f002:**
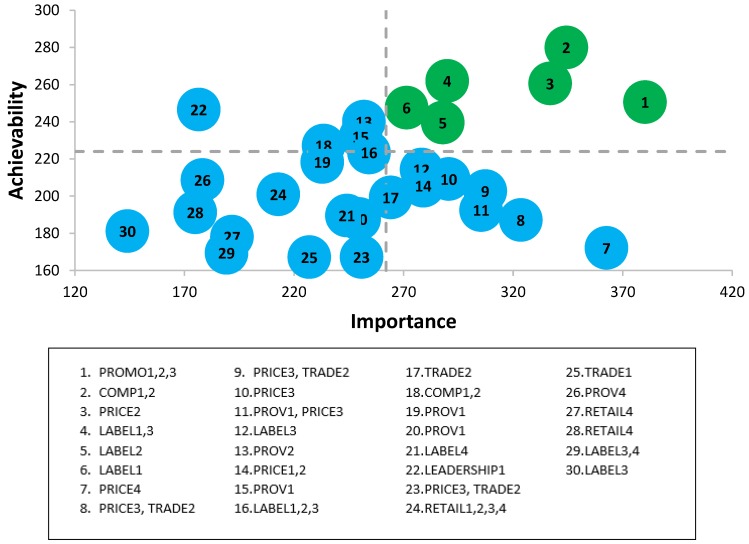
Importance and achievability of policy actions for the Canadian federal government according to expert scores, and the Food-EPI domains which the policy falls within, Food-EPI Canada, 2017. Detailed descriptions of proposed policy actions can be found in Table 1.

**Figure 3 ijerph-16-04473-f003:**
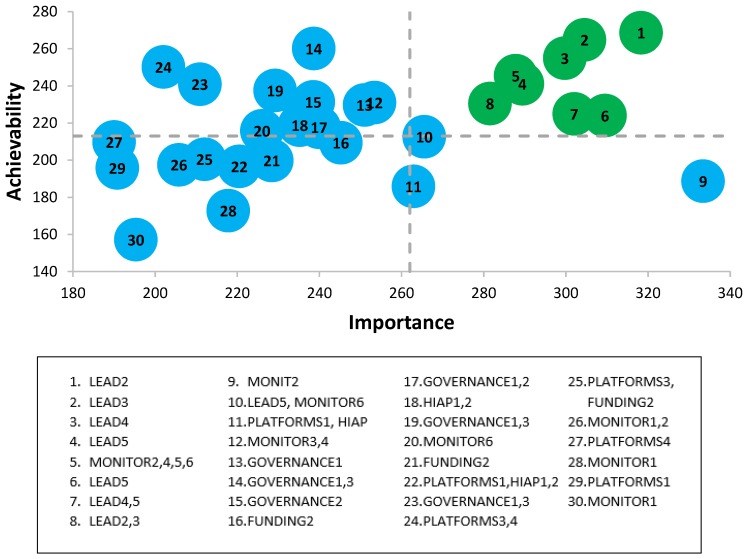
Importance and achievability of infrastructure support actions for the Canadian federal government according to expert scores, and the Food-EPI domains which the policy falls within, Food-EPI Canada, 2017. Detailed descriptions of proposed policy actions can be found in Table 2.

**Table 1 ijerph-16-04473-t001:** Prioritized list of proposed policy actions for the federal government to take, according to expert ratings of importance and achievability of the policy actions, Food-EPI Canada, 2017.

1. Implement a comprehensive federal policy to prohibit advertising of unhealthy foods and beverages as identified by a comprehensive, evidence-based nutrient profiling system to children under the age of 17 through all forms of media that are or can be targeted to children in this age group (including broadcast media, non-broadcast media, and in children’s settings), with an eventual goal to prohibit all advertising to children.
2. Implement targets for sodium, free sugar, and saturated fat in the food supply (packaged foods and restaurant foods) using a structured voluntary approach with the threat of mandatory requirements if compliance is poor after an established time period, and implement a monitoring system for nutrients of concern in the food supply to track compliance.
3. Implement an excise tax on all sugary drinks, as defined by comprehensive, evidence-based nutrient profiling criteria. Invest the revenue from the tax to targeted areas that address health as appropriate to context, and advertise the re-investment of the tax dollars to the public.
4. Include a declaration and %DV for free sugar content in the Nutrition Facts table and in front-of-package labelling schemes.
5. Develop clear and consistent nutritional criteria that must be met in order for any food or drink products to carry a health claim or nutrient content claim; any foods that are high in nutrients of concern would not be permitted to carry a claim.
6. Extend the current Nutrition Facts table requirements to require nutrition labelling be applied to centrally prepared, in-store take away foods, baked goods and pastries, packaged sandwiches and salads, meat products, baby foods, and alcohol, and require websites to display Nutrition Facts tables for any packaged foods that are sold online that carry a Nutrition Facts table on their package.
7. Implement a national minimum basic income for all people living in Canada, to enable all to afford healthy food, as part of a comprehensive Poverty Reduction Strategy for Canada.
8. Develop agricultural policies and subsidies that incentivize production, processing, distribution, and consumption of unprocessed or minimally processed vegetables, fruit, and legumes that are healthy, local, and sustainable.
9. Provide federal subsidies to increase local capacity for food production and innovation and community-based health promotion interventions in the provincial Norths and far North to address food sovereignty issues.
10. Continue to increase the scale and scope of programs and strategies (such as Nutrition North Canada) to provide improved equitable access to affordable, healthy food among Canada’s provincial Norths and far North.
11. Provide federal funding and support for (1) a national school feeding program and (2) fruit and vegetable programs to be implemented by provinces and territories in schools both on and off reserve.
12. Implement a standardized, comprehensive labelling system on the front of packaged foods and on restaurant foods that has been developed using a comprehensive, evidence-based nutrient profiling system to provide consumers with simple, interpretive information on the healthfulness of products at the point of sale.
13. Develop and implement clear, consistent policies including public procurement standards to provide and promote healthy food and beverage choices in food service activities (cafeterias, vending machines, food at events, fundraising, promotions, etc.) in settings under federal government control (government buildings, national parks) using a coordinated approach.
14. Alter GST/HST regulations such that exemptions from GST are considered based upon the healthfulness of food and beverage products, using a comprehensive, evidence-based nutrient profiling system, and complement this with a public awareness campaign to inform consumers of the changes to sales tax regulations in grocery stores.
15. Develop available resources and provide technical assistance and guidance to the provinces and territories to develop and implement healthy nutrition standards in schools and early childhood education settings and other publicly-funded settings.
16. Implement consistent and ongoing monitoring and enforcement of nutrition labelling, and ensure that the system and results are transparent and available to the public.
17. Ensure that specific and explicit provisions are included in trade and investment agreements that allow the government to preserve its regulatory capacity to protect and promote public health nutrition.
18. Use a structured voluntary approach to set portion size standards for both packaged and restaurant foods in line with dietary guidance.
19. Establish Federal/Provincial/Territorial guidelines for foods that are permitted to be provided or sold in early childhood education settings.
20. Require mandatory implementation of the Federal/Provincial/Territorial guidelines for foods sold or provided in schools and similar consistent guidelines for early childhood education centers as a minimum that is required to be legislated by provinces and territories.
21. Implement a national menu labelling policy with calorie and sodium information on menus and menu boards for all chain food service establishments nationwide with a comprehensive menu labelling education campaign and added fiscal incentive for industries. In addition, require chain food service establishments to fully disclose amounts of energy and the 13 core nutrients found on the Nutrition Facts table per serving size for foods and beverages in an online format.
22. Provide vocal support to the World Health Organization to strengthen recommendations for public health nutrition.
23. Implement targeted commodity subsidies and subsidized transportation for vegetable, fruit, and legume producers that support local and sustainable production to reduce costs in domestic markets and increase consumption.
24. Develop national guidelines to support provincial/territorial authorities to develop supplementary planning guidance and simplified mechanisms within planning laws to enable the development of policies to promote and equitably access healthier food options and/or discourage less healthy food options at the local level.
25. Include formal and explicit public health nutrition and health risk assessments as part of national interest analysis on trade and investment agreements, and include considerations regarding the economic burden of diet-related NCDs in trade and investment analyses.
26. Develop programs and support to stimulate and incentivize industry sectors and large employers to create pledges for healthier food environments in the workplace.
27. Restrict offers on unlimited sugary drinks for free or at fixed prices in restaurants.
28. Establish a mechanism to provide synthesized, evidence-based guidance and support for retailers and food service outlets to both encourage and enable them to provide healthier food choices.
29. Explore opportunities to extend front-of-package (FOP) labelling policy implemented for packaged foods to restaurant foods to provide additional information on nutrients of concern at the point of sale.
30. Explore interactive platforms to clearly communicate interpretive front-of-package information to consumers (using tablets and smartphones).

**Table 2 ijerph-16-04473-t002:** Prioritized list of proposed infrastructure support actions for the federal government to take, according to expert ratings of importance and achievability of the infrastructure support actions, Food-EPI Canada, 2017.

1. Develop public targets for intake of sodium, saturated fat, and free sugar, and vegetables and fruit, monitor progress and inequities in achieving targets over time.
2. Implement evidenced-informed revisions for a more comprehensive, multi-component Canada’s Food Guide, with recommendations for the public that promote a holistic view of the food environment and system and acknowledge environmental impact, sustainability, and cultural appropriateness, in addition to a practical resource for practitioners and policy makers that includes a nutrient- and food-specific approach to public health.
3. Monitor and revise the Healthy Eating Strategy on an on-going basis with a timeline for policy implementation and action, and establish a long-term vision for the Healthy Eating Strategy with a timeline that extends beyond the current election cycle, which includes a research agenda and evaluation plan that is adequately resourced and aligns with the objectives and outcomes of the Strategy.
4. Include specific actions and policies in the Healthy Eating Strategy and other food-related policies to improve population nutrition among vulnerable and disadvantaged groups, with a broad definition of vulnerable groups (e.g., those with low socio-economic status, children and Indigenous peoples, among others exhibiting social and health disparities).
5. Mandate the collection of food security data in the Canadian Community Health Survey across all provinces and territories to comprehensively understand the state of food insecurity across the country at a national and provincial/territorial level on an ongoing basis.
6. Establish a national Poverty Reduction Strategy that includes a specific focus on reducing household and community food insecurity and nutrition-related health inequities.
7. Establish a comprehensive food and agriculture policy for Canada with an increased focus on public health nutrition, environmental sustainability, and local food production with strong collaboration across sectors and between federal and provincial/territorial governments that aligns with the Healthy Eating Strategy and is particularly mindful of vulnerable groups.
8. Establish a comprehensive nutrient profiling system, in alignment with national dietary guidance, that can be used and adapted across policy interventions to simplify implementation, maintenance, and monitoring across policies.
9. Comprehensively assess dietary intake among the entire population including vulnerable populations and those living in rural and remote areas at least every 5 years, working with representatives from each province and territory, and incorporate specific biomarkers and evidence-based anthropometric measurements to be collected alongside dietary intake data among a subset of the sample.
10. Establish measurable goals to identify and close the gaps in health outcomes between Aboriginal and non-Aboriginal communities, and to publish annual progress reports and assess long-term trends, as recommended in the Truth and Reconciliation Commission Calls to Action.
11. Create a whole-of-government approach at the federal and Federal/Provincial/Territorial level around nutrition issues.
12. Increase accessibility and capacity to use provincial/territorial administrative databases by researchers to monitor health-related risk factors and outcomes, such as NCD rates.
13. Set clear guidelines for involvement of different stakeholders in policy development processes, ensuring that food industry representatives are not involved in setting policy objectives and agendas where they have conflicts of interest with improving population nutrition.
14. Expand implementation of the transparency policy being applied to the Healthy Eating Strategy to the development or revision of all food policies.
15. Establish requirements and a process for the collection and use of evidence in all federal food policies.
16. Establish a long term and timely funding initiative for population health intervention research and surveillance relating to diet-related non-communicable diseases.
17. Regulate the disclosure of the amount and type of funding support provided by any industry (including the food industry and commodity sectors) to all researchers.
18. Require formal health impact assessments as part of food and non-food policy development and proposal processes, including explicit details about the consideration of potential impacts of policies on population nutrition and health.
19. Publicly post the amount of money spent by industry on lobbying the federal government as part of the Lobbying Act and the Lobbying Registry Act.
20. Publish reports related to progress on the reduction of health inequities using available, nationally representative data.
21. Ensure research funding allocation takes into account the diseases and conditions with the highest burden, including an increase in the proportion of research funding that is allocated specifically to improving population nutrition and diet-related NCD prevention research.
22. Implement a nutrition-sensitive program for agriculture in Canada to support and promote a food system that aligns with public health nutrition goals.
23. Post all comments submitted to policy consultations and regulatory changes publicly, as is done in the US Dockets system.
24. Develop capacity among civil society groups and organizations to participate in policy consultation and development, including provision of appropriate platforms and resources to allow groups to participate fully in consultations and policy development, and ensuring accessibility of policy conversations by addressing language barriers, literacy, and access to technology.
25. Increase the opportunity for policy-maker and researcher partnerships and provide infrastructure support for the development, monitoring, and evaluation of government policies.
26. Improve Public Health Agency of Canada’s policy and surveillance capacity for public health nutrition.
27. Provide support for local, community-based interventions to create healthy food environments by implementing and funding a knowledge translation and exchange platform to provide advice, promote best practice, and offer networking opportunities, such as an up-to-date best practices portal.
28. Establish an ongoing and publicly-disclosed monitoring system for the nutritional quality of the food supply, foods served in school and early childhood education environments, food marketing to children, and retail food environments.
29. Establish an ongoing platform for improved communication and relationships between Health Canada and Agriculture and Agri-Food Canada.
30. Establish monitoring of traditional country food and water supply, food availability, and measures of food sovereignty in rural and remote areas.

**Table 3 ijerph-16-04473-t003:** Policy and infrastructure support actions that were prioritized as important and achievable across many or most provinces and territories.

**Policy Actions**
Provide a universal fruit and vegetable program in all schools.
Strengthen school nutrition standards, and harmonize these standards to have a consistent guideline for foods served in settings where children gather, including early childhood education centers, and other public sector settings frequented by children.
Strengthen nutrition standards and procurement policies for public sector settings and provide support for successful implementation of these policies, including long term care facilities, hospitals, recreation centers, and correctional services, etc.
Examine current school curricula with regards to food literacy, and introduce food literacy and food skills training as a mandatory component of school curricula.
Introduce a comprehensive provincial or territorial policy restricting marketing of unhealthy food and beverages as identified by a comprehensive, evidence-based nutrient profiling system to children under the age of 17 in public settings frequented by children, such as arenas and community centers with a sufficient enforcement mechanism to ensure compliance.
Implement a point-of-sale sales tax on all sugary drinks, as defined by comprehensive, evidence-based nutrient profiling criteria. Invest the revenue from the tax to targeted areas that address public health as appropriate to provincial context, and advertise the re-investment of the tax dollars to the public.
**Infrastructure Support Actions**
Establish an up-to-date strategy or framework for public health nutrition and healthy eating with comprehensive food environment considerations, and specific actions and policies in the health strategy to improve population nutrition among vulnerable and disadvantaged populations.
Acknowledge and endorse the importance of public health nutrition and obesity and non-communicable disease prevention strategies in political platforms, mandate letters and speeches from the throne.
Work with First Nations/ Inuit/ Metis leadership and others to develop a comprehensive strategy to promote access, availability, and affordability of healthy foods for Indigenous populations on and off reserve within the context of local foodways and cultural traditions.
Establish measurable goals to identify and close the gaps in health outcomes between Aboriginal and non-Aboriginal communities, and publish annual progress reports and assess long-term trends, as recommended in the Truth and Reconciliation Commission Calls to Action.
Establish a Healthy Eating Committee that includes representation from all sectors (government, private sector, and civil society) with sufficient resources to support participation of non-governmental groups.

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
