# Peer review of "Policies to Create Healthier Food Environments in Canada: Experts’ Evaluation and Prioritized Actions Using the Healthy Food Environment Policy Index (Food-EPI)"

_ijerph, 2019, doi:10.3390/ijerph16224473_

Round 1
Reviewer 1 Report
The work presented in this paper is extremly interesting and applicable and shows an accurate methodology, I have only some minor changes to notice
dots and comas should be written after citations not before line 190: ....Federal Evidence document (1=not sufficientlycomprehensive, 5=too comprehensive).... can something be "TOO comprehensive" is this the best term to use??? references 2/20/21/22/24/25/26/28/29/30/36 lack date of accessit reference 13: pages are missed
Author Response
RESPONSE TO REVIEWER 1:
The work presented in this paper is extremely interesting and applicable and shows an accurate methodology, I have only some minor changes to notice
Dots and comas should be written after citations not beforeThese changes have been made.
Line 190: ....Federal Evidence document (1=not sufficiently comprehensive, 5=too comprehensive).... can something be "TOO comprehensive" is this the best term to use???
While we appreciate that this is somewhat odd phrasing, we believe that the participants understood the question. The question aimed to understand if participants thought there were too many details or too much information such that it was difficult to understand the policy. At this point, we are unable to make any changes as this was the actual measure that was asked of the participants. We will consider this for future research.
references 2/20/21/22/24/25/26/28/29/30/36 lack date of access
These references have been corrected so the date is added.
reference 13: pages are missed
This reference has been corrected.

Reviewer 2 Report
This is a well-written and informative manuscript that touches on a topic of global importance - the relationship between food environments and healthy eating habits and the role of policy in supporting transition to healthier eating habits.
There were a few minor errors in the manuscript, some caused by referencing, which I have highlighted in the manuscript.
My main comment is to strengthen the discussion by citing more global literature, comparing the study's findings to the literature, and providing implications of the study's findings. While this was a Canadian study, it is important to be able to still link to the global picture on nutrition so that lessons can be drawn and applied elsewhere. That would broaden the applicability of the study's findings.
I would also recommend adding a section on limitations before the Conclusions, to put into perspective some of the study's limitations that are relevant to intepretation of the results. I saw some mentioned in the Methodology, but not in the Discussion.
Author Response
RESPONSE TO REVIEWER 2:
This is a well-written and informative manuscript that touches on a topic of global importance - the relationship between food environments and healthy eating habits and the role of policy in supporting transition to healthier eating habits.
There were a few minor errors in the manuscript, some caused by referencing, which I have highlighted in the manuscript.
We would like to thank the reviewer for their detailed review. We have gone through the manuscript and extracted comments and highlighted areas and responded below.
THROUGHOUT – We have fixed the references so that the punctuation comes after the brackets.
Line 39 – we have changed this to sodium intake.
Line 41 – we have changed this to read “Globally, food environments are currently not conducive to supporting healthy eating habits [1,4,10] and Canada is no exception.” We believe that the following text is the evidence that demonstrates how the Canadian environment is not supportive of healthy eating.
Line 69 – We have changed the word ‘elsewhere’ to ‘in previous publications’ in this instance and on line 293.
Line 94 – As the result of a computer programming error, the indicator was not rated by participants. Additional clarification has been added.
Line 104 – This link will be permanently available.
Line 117 – We have added several references for interested readers to better understand the Nunavut food environment if they so choose.
Line 263 – The Figure caption has been changed to:
Figure 1. Composite score of federal and provincial/territorial ratings of the level of implementation compared to best practice for 46 Food-EPI indicators within 7 policy and 6 infrastructure support domains.
Line 379 – We have added some additional context on these policies being the needed, impactful, feasible and achievable, according to the expert ratings.
“The policies that were prioritized were identified as most likely to fill the greatest gaps in the current environment and have the greatest impact, while still having high acceptability and feasibility. It is likely that the importance and achievability of some of these policies will change over time, as other policies are introduced, sociocultural norms and behaviours change, and political priorities shift. “
Line 410 – We have named the prioritized policy and infrastructure support actions in the final paragraph .
My main comment is to strengthen the discussion by citing more global literature, comparing the study's findings to the literature, and providing implications of the study's findings. While this was a Canadian study, it is important to be able to still link to the global picture on nutrition so that lessons can be drawn and applied elsewhere. That would broaden the applicability of the study's findings.
We have added to the paragraph in the discussion to broaden the implications of the study. In particular, we have added several sentences stating:
Notably, no countries received a ‘high’ Food-EPI score, indicating that globally, few countries are implementing comprehensive policies and infrastructure supports the enable healthier food environments and support healthier food choices [41] Several of the prioritized policy and infrastructure support actions highlighted in Food-EPI Canada have also been highlighted by other countries as being the most important and achievable actions that governments can consider, in particular the recommendations to restricting advertising of unhealthy foods to children and implement a sugary drink tax. More than 40 jurisdictions in over 20 countries have implemented sugary drink taxes, and at least 8 countries have mandatory restrictions on the advertising of unhealthy foods to children via broadcast or non-broadcast media [21]. This demonstrates the policy momentum that is gaining for some of these policies that were once considered radical [42,43].
I would also recommend adding a section on limitations before the Conclusions, to put into perspective some of the study's limitations that are relevant to interpretation of the results. I saw some mentioned in the Methodology, but not in the Discussion.
We have outlined strengths and limitations of the paper, but perhaps this wasn’t clear. We have added a heading so that this paragraph is now highlighted. In addition, we have added a sentence that states:
“Lastly, the scope of the Food-EPI process and tool at present is limited to nutrition issues related to obesity and non-communicable diseases, and does not assess other policy elements that are of interest to policymakers, such as environmental and climate change-related issues. Future work may consider including more broad aspects of the food system to identify synergistic policy opportunities.” (Line 371)
Round 2
Reviewer 2 Report
The authors have addressed and responded to the comments that were made in the first manuscript. I am satisfied with the responses provided.